# The Effect of Isolated and Combined Application of Menthol and Carbohydrate Mouth Rinses on 40 km Time Trial Performance, Physiological and Perceptual Measures in the Heat

**DOI:** 10.3390/nu13124309

**Published:** 2021-11-29

**Authors:** Russ Best, Seana Crosby, Nicolas Berger, Kerin McDonald

**Affiliations:** 1Centre for Sports Science and Human Performance, Waikato Institute of Technology, Hamilton 3200, New Zealand; seacro02@student.wintec.ac.nz (S.C.); Kerin.McDonald@wintec.ac.nz (K.M.); 2School of Health and Life Sciences, Teesside University, Middlesbrough TS1 3BX, UK; N.Berger@tees.ac.uk

**Keywords:** menthol, carbohydrate, cycling, time trial, endurance, heat

## Abstract

The current study compared mouth swills containing carbohydrate (CHO), menthol (MEN) or a combination (BOTH) on 40 km cycling time trial (TT) performance in the heat (32 °C, 40% humidity, 1000 W radiant load) and investigates associated physiological (rectal temperature (Trec), heart rate (HR)) and subjective measures (thermal comfort (TC), thermal sensation (TS), thirst, oral cooling (OC) and RPE (legs and lungs)). Eight recreationally trained male cyclists (32 ± 9 y; height: 180.9 ± 7.0 cm; weight: 76.3 ± 10.4 kg) completed familiarisation and three experimental trials, swilling either MEN, CHO or BOTH at 10 km intervals (5, 15, 25, 35 km). The 40 km TT performance did not differ significantly between conditions (F_2,14_ = 0.343; *p* = 0.715; η^2^ = 0.047), yet post-hoc testing indicated small differences between MEN and CHO (d = 0.225) and MEN and BOTH (d = 0.275). Subjective measures (TC, TS, RPE) were significantly affected by distance but showed no significant differences between solutions. Within-subject analysis found significant interactions between solution and location upon OC intensity (F_28,196_ = 2.577; *p* < 0.001; η^2^ = 0.269). While solutions containing MEN resulted in a greater sensation of OC, solutions containing CHO experienced small improvements in TT performance. Stimulation of central CHO pathways during self-paced cycling TT in the heat may be of more importance to performance than perceptual cooling interventions. However, no detrimental effects are seen when interventions are combined.

## 1. Introduction

The individual cycling time trial (TT), commonly referred to as “the race of truth”, has long been a test of an individual’s cardiovascular capacity, muscular endurance and fortitude [1]. Elite cyclists complete 40–50 km distances, averaging speeds in excess of 48 kph [2]. Studies in trained cyclists have shown average wattage (W) to be greater than 300 W in both indoor and outdoor settings [3], with heart rate (HR) averaging between 85 and 90% of maximum [4]. The physiological cost of this gruelling event has been compounded in recent Olympic cycles due to the added environmental stresses encountered by athletes. To illustrate this, Pfeiffer and Abbiss [5] assessed 40 km TT performance in hot and cool conditions (32 °C and 17 °C, respectively). Heat adversely affected mean power output (W), heart rate (HR), core temperature (T_rec_), rating of perceived exertion (RPE) and thermal sensation (TS) and were significantly different from measures obtained in cool conditions. As such, sports scientists, coaches and athletes continue to look at ways to improve performance when competing in the heat.

Heat acclimation protocols and pre-event cooling strategies have been well documented in aiding performance in the heat [6,7]. Both interventions allow physiological changes that aid in the reduction of core temperature and influence subjective measures of thermal comfort and sensation. An emerging area of interest is the stimulation of gustatory and thermo-receptors within the oral cavity, due to potential physical and psychological benefits whilst exercising [8]. Activation of these receptors can be achieved via simply swilling a liquid solution within the mouth before expelling or ingesting it.

Research into the effect of swilling carbohydrate (CHO) and menthol (MEN) solutions prior to and during exercise is now established. Both tastants have been shown to have an ergogenic effect without requiring ingestion [9,10,11,12]. The level of performance enhancement may vary depending on type and duration of exercise [11], timing, concentration and length of the swill [13]. While the mechanisms responsible still require some elucidation, it is thought that stimulation of oral receptors may result in central nervous system activation, causing physiological and psychological responses [8,14].

More specifically, research pertaining to CHO swilling indicates stimulation of type 1, member 2 and 3 taste receptors (T1R2, T1R3) on the tongue [15] cause reward centres within the brain to be activated through dopamine pathways [16]. It is possible that this activation of reward centres leads to increased motivation and subsequent increases in exercise intensity. Reviews quantifying the effect of CHO mouth swill have found that the greatest possible effects are experienced during endurance exercise lasting 25–60 min, with smaller effect sizes reported for resistance, anaerobic and shorter duration exercise [11]. Cycling TTs with CHO swilling have resulted in significantly higher power outputs compared to placebo but not for time to completion [17]. Performance is typically improved to a greater extent when conducted in a fasted state but would still be considered nutritionally optimised in the fed state. While effect sizes are small across studies, they may be considered practically worthwhile in elite populations [11,18].

Prior nutritional state may also influence levels of improvement. Swilling a CHO solution at regular time points significantly improved mean power output in both fed and fasted states during 1 h cycling TT compared to placebo [19]. The duration of the swill may also be a factor, with prolonged exposure potentially causing enhanced stimulation of receptors and improvements in performance [13]. The practicality of a longer swill time is contentious, as it may not be viable for athletes to undertake this strategy during high-intensity exercise. Furthermore, it is less clear if performance improvements are observed in the heat, e.g., Cramer et al. [20] found no significant difference between repeated CHO swilling and placebo during 40 km TT in the heat. It is possible that thermal strain may have mitigated any performance enhancement due to elevated levels of physiological stress (e.g., increased cardiac drift and T_rec_) and worsened subjective ratings of thermal sensation and thermal comfort [20].

A potential strategy to improve subjective measures in the heat is using a menthol mouth swill. The minty flavour imparted from menthol is experienced in many products, including chewing gum, topical analgesics and dental products. Menthol stimulates the voltage-gated ion channel, transient receptor potential melastatin 8 (TRP-M8), and in doing so mimics the feeling of cold–cool temperatures (8–28 °C) [21]. This has led researchers to investigate the possible ergogenic effects menthol may have, particularly when exercising in the heat.

A review by Gavel et al. [10] found 8 of 10 studies utilising a MEN mouth swill improved performance across a variety of protocols (concentrations, durations, environmental conditions, swill frequency, sex). Specifically in cycling time trials, menthol mouth swill significantly improved mean power output (*p* = 0.034) and 30 km TT performance (2.3%) [22]. Beverages of varying temperatures that contained menthol also significantly improved 20 km TT compared to those without (*p* = 0.02) [23]. External menthol sprays, however, did not significantly improve 40 km cycling performance [24]. Not all exercise modalities or thermal conditions may benefit from menthol swilling. For example, Gibson et al. [25] showed no improvement in repeated sprint ability in the heat, and Best et al. [26] found no significant difference in strength and power activities in thermal neutral conditions. For activities and environmental conditions where heat stress is minimal, it is likely that menthol would only induce ventilatory responses, as the scope for influence on thermal perception is negligible.

When swilled, menthol stimulates thermal receptors in the mouth, via the trigeminal nerve, imparting the sensation that the body is experiencing colder temperatures [21]. In cooler conditions, this may be of less benefit. However, it is generally proposed that if an athlete feels cooler and more comfortable in hot conditions, they may exercise at a higher intensity and or for longer, as per a recent meta-analysis of menthol swilling during exercise in the heat [9].

As previously mentioned, CHO swilling has been shown to improve performance [11], but not in the heat [20], whereas MEN improves perceptions of cooling, facilitating performance in hot conditions. Best et al., [27] found menthol improved thermal comfort and thermal sensation in the heat by small (*d* = −0.33) and moderate (*d* = −0.71) levels, respectively, compared to a CHO swill. As such, it is postulated that including a menthol mouth swill may help to improve thermal subjective measures while exercising, whilst allowing signalling of CHO pathways to be maintained. Therefore, the current study aims to assess the effects of isolated and combined application of MEN and CHO solutions on 40 km TT performance in the heat. Further assessment of physiological and subjective measures will indicate whether an additive effect has been observed. It is hypothesised that when perceptual cooling (MEN) and CHO swilling are provided at regular intervals concomitantly, performance may be enhanced compared to MEN or CHO swilling individually.

A secondary aim of the study was to investigate the location and intensity of oral cooling induced by menthol, compared to placebo (peppermint in CHO) and when combined with CHO. The use of perceptual measures in MEN mouth swill studies is common; however, thermal sensation may be location specific but is reported as an integrated perception. It is possible that the impact and duration of time MEN influences thermal sensation may differ orally to a gestalt interpretation. It is unclear at present whether oral cooling persists when MEN and CHO solutions are combined; whilst the substances target differing pathways, one tastant may supersede the other, dependent upon the dominant physiological signalling induced by the environmental conditions and exercise demand. The above research aim and hypothesised mechanisms of performance enhancement are summarised in Figure 1 below.

## 2. Materials and Methods

A randomised, double-blind, repeated-measures design was employed, with trial order assigned by a Latin square following familiarisation with no intervention; there was a washout period of ≥7 days between trials. Ethical approval for this study was obtained from the Human Ethics Research Group, Wintec (Approval Number: WTFE07110419).

### 2.1. Participants

All participants were non-heat acclimated at the start of the study, with testing taking place in winter. Ten male participants (age: 32 ± 9 y; height: 180.9 ± 7.0 cm; weight: 76.3 ± 10.4 kg) were recruited for this study, of the 10, eight completed all trials (age: 31.4 ± 9.1 y; height: 179.5 ± 6.9 cm; weight: 75.8 ± 11.0 kg); two participant withdrew due to injuries not related to participation in the study and were excluded due to insufficient data.

### 2.2. Experimental Solutions

All solutions were provided at 10 km intervals from 5 km onwards (5 km, 15 km, 25 km, 35 km) in 25 mL aliquots. Menthol was provided at a 0.1% concentration, dissolved from crystals (manufacturer’s details) and diluted as per Best et al. [28]. Carbohydrate swills contained 10% maltodextrin (Equagold Tapioca Maltodextrin, Equagold, Takaanini, Auckland, New Zealand), with non-menthol-containing peppermint flavouring added to aid participant blinding. The combined swill (menthol and carbohydrate) was made by adding carbohydrate to the menthol solution. All solutions were coloured light green to maximise sensory expectancy [26,28] and facilitate blinding of participants and researchers.

### 2.3. Procedure

Prior to trial commencement, participants were provided with a rectal thermistor compatible with a data logger (Squirrel Series 10, Grant Instruments, Cambridge, England) to self-insert and urine collection pottle. Mid-stream urine samples were provided for the assessment of specific gravity, which was used as a marker of hydration status (<1.020). If athletes presented a specific gravity >1.020, 500 mL of water was consumed over a 20 min period [29]. Participants and their water bottles were then weighed (Seca 869, Seca, Birmingham, UK); change in body mass and bottle weight was considered dependent variables. Upon entering the environmental chamber, participants completed a 10 min warm up at a self-selected intensity. The Wattbike^™^ (Wattbike Pro; Wattbike; Nottingham, UK) was positioned with the anterior of the bike 1.2 m from the front wall and fan(s) and 2 m from the posterior wall, approximately central. The radiant heat source (2000 W; Trade Tested Limited, Auckland, New Zealand) was positioned at a height of 2 m on the posterior wall and emitted heat onto the back of the participant, angled at 45°, and provided 1000 W of radiant heat load. This positioning was chosen in an attempt to replicate ecologically valid conditions by targeting large heat-sensitive areas on the anterior and posterior of the athlete, for convective cooling (fan) and radiant heating, respectively. Environmental conditions were set at 32 °C and 40% humidity, and participants were asked to wear a cycling helmet of their choosing during exercise.

Following completion of their warm up, participants were asked to complete a 40 km TT in the shortest possible time. During the TT, participants could see the distance remaining within 5 km splits, but they were blinded to all other information. Four swills were performed at 10 km intervals starting from 5 km (5, 15, 25 and 35 km); swills were provided in 25 mL aliquots and performed for 8 s, prior to expectoration. During each TT attempt, the time taken to complete each 5 km split and the 40 km TT was recorded. Average and peak absolute power (W) and average relative power (W/kg) per 5 km split were recorded; similarly, T_rec_ and HR were recorded throughout. Subjective measures of TS, TC and differential ratings of perceived exertion for participants’ legs (peripheral) and lungs (central) were gathered at 5 km intervals, using scales by Zhang [30] and the Borg CR10 scale, respectively [31]. Thirst was gathered at 5 km intervals, rated between 0 “Not at all thirsty” to 9 “Severely thirsty” [32]. When swilling, thirst was noted pre- and post-swill to capture the effect of swilling and treatment concomitantly. Perceived location and intensity of oral cooling were also measured using a novel scale, whose scale and descriptors aligned to those for TC and TS (0, No cooling; 1, Weak; 2, Moderate; 3, Strong; 4, Very Strong). Participants could rate the intensity of oral cooling across the forehead, nasal and oral cavities, throat and upper respiratory region. This is to capture regions that are sensitive to menthol either through trigeminal innervation or tissues that are sensitive to menthol due to the thickness of their strata cornea. Pilot work had suggested that sensations of cooling induced by menthol may be experienced differently over time. A blank version of this scale is available below (Figure 2) and in the Appendix A. Water was available ad libitum throughout the trial, with total volume ingested (mL) also considered a dependent variable.

### 2.4. Statistical Analyses

Prior to specific analyses, data were assessed for normality and homogeneity of variance, against previously detailed criteria [33,34]. Sample size was not established using statistical power but was limited by resources, influenced by pandemic protocols and the access to trained athletes late in an Olympic cycle [35]. Analyses were performed in JASP (JASP team (2020) v0.14.1; University of Amsterdam). Where Maulchy’s test of sphericity was significant, a Greenhouse–Geisser correction was applied and used for subsequent interpretation of effects. Repeated contrasts were used to examine the effects of distance and distance x solution interactions, simple contrasts were employed to assess between solution differences. Due to the large number of multiple comparisons being performed, there is a risk of type I error, which is minimised by applying a Bonferroni correction in *post-hoc* analyses. Where appropriate, Cohen’s *d* was used as a further measure of effect size, as this better reflects pairwise comparisons than eta squared (η^2^), which is reported alongside main ANOVA statistics and interpreted as follows: *small*, 0.01; *medium*, 0.06; *large*, ≥0.14.

The 40 km TT performance (time) was assessed via a one-way repeated-measures ANOVA and followed up with equivalence testing, using solution pairs and a raw equivalence range of −1.34 to 1.34, corresponding to ±80.4 s or 2% variation in performance [36]. Performance, physiological and perceptual variables, with the exception of thirst, were assessed via a two-way repeated-measures ANOVA, with solution and distance completed as within-subject factors. Thirst was converted into a change score between pre- and post-swilling values prior to undergoing the same analysis. Location and intensity of oral cooling were assessed using a three-way repeated-measures ANOVA, with solution, distance completed and location as within-subject factors. This approach has previously been used to detect differences between regional thermal sensations [37]. Statistical significance was set at *p* < 0.05, readers are also encouraged to interpret these data alongside typical and ecologically meaningful variation in performance and other measures, e.g., the smallest worthwhile change of all perceptual measures was considered 0.5 arbitrary units [38,39].

## 3. Results

### 3.1. 40 km Time Trial Performance

The 40 km TT performance did not differ significantly between familiarisation and experimental conditions (*F*_3,18_ = 1.903; *p* = 0.165; η^2^ = 0.241). However, post-hoc testing showed *moderate* differences between familiarisation and all other trials (*d* = 0.637−0.796; *p* = 0.297−1.000)

The 40 km TT performance did not differ significantly between experimental conditions (*F*_2,14_ = 0.343; *p* = 0.715; η^2^ = 0.047; Figure 3 Panel A). However, *post-hoc* testing indicated *small* differences between solutions MEN and CHO (*d* = 0.225) and MEN and BOTH (*d* = 0.275); this is further supported by equivalence testing, whereby the upper bounds of these comparisons were deemed to not be equivalent t(7.00) = −1.029, *p* = 0.169 and t(7.00) = −1.308, *p* = 0.116. When conditions CHO and BOTH were compared though, they were deemed to be statistically equivalent, t(7.00) = −2.228, *p* = 0.031, indicating that the true difference lies within ±84 s when the effects of carbohydrate-containing solutions are compared.

Mean power output did not differ significantly between trials (*F*_2,14_ = 0.103; *p* = 0.903; η^2^ = 0.014; Figure 3 Panel B). When examined in 5 km segments, power output (W) was significantly affected by distance (*F*_7,49_ = 3.352; *p* = 0.005; η^2^ = 0.197) but was not influenced by solution (*F*_2,14_ = 0.103; *p* = 0.903; η^2^ = 0.003). Mean power for the penultimate (30 km–35 km) and final (35 km–40 km) segments differed significantly for all solutions, indicating a marked end spurt. More specifically solutions CHO and BOTH demonstrated *large* differences in power output (*p* < 0.001; *d* = 2.19; 90% CI 1.04 to 3.25 and *p* = 0.008; *d* = 1.30; 90% CI 0.46 to 2.07, respectively), and Solution A showed *very large* differences between these intervals (*p* < 0.001; *d* = 2.98; 90% CI 1.52 to 4.34).

### 3.2. Physiological Measures

Sweat loss throughout trials did not differ between solutions (*F*_3,15_ = 1.552; *p* = 0.242; η^2^ = 0.237), ranging from 1.48 L to 1.64 L. Fluid ingested did not differ significantly between solutions (*F*_2,14_ = 0.849; *p* = 0.449; η^2^ = 0.108); however, MEN demonstrated *small* reductions in fluid intake compared to CHO (*d* = −0.46; −0.24 L; 90% CI −0.67 L to 0.19 L) and BOTH (*d* = −0.252; −0.13 L; 90% CI −0.56 L to 0.30 L). On the other hand, CHO induced a *small* increase in fluid consumption compared to BOTH (*d* = 0.208; 0.11 L; 90% CI −0.32 L to 0.54 L).

T_rec_ was significantly affected by distance (*F*_9,63_ = 35.70; *p <* 0.001; η^2^ = 0.553; Figure 3 Panel C), but there was no significant time x solution interaction (*F*_18,126_ = 1.232; *p =* 0.246; η^2^ = 0.032), or independent effect of solution, within subjects (*F*_2,14_ = 3.215; *p =* 0.071; η^2^ = 0.039). When expressed as a rate of change (T_rec_ Δ), there was a significant effect of time (*F*_8,32_ = 2.587; *p* = 0.027; η^2^ = 0.111), with a significant increase noted between 5 km and 10 km (*p* = 0.025).

HR was also significantly affected by distance (*F*_9,63_ = 134.65; *p <* 0.001; η^2^ = 0.892; Figure 3 Panel D), but there was no significant time x solution interaction (*F*_18,126_ = 0.850; *p =* 0.638; η^2^ = 0.005), or independent effect of solution, within subjects (*F*_2,14_ = 3.106; *p =* 0.076; η^2^ = 0.004). More specifically, HR increased between entry and warm up, warm up completion and 5 km (both *p* < 0.001) and 5 km–10 km (*p* = 0.015). HR then remained stable until the final 5 km interval, where it again increased significantly (35 km–40 km; *p* = 0.003). Similar to T_rec_ Δ, HRΔ increased between 5 km and 10 km (*p* < 0.001) and between 35 km and 40 km (*p* = 0.058).

### 3.3. Perceptual Measures

TS was significantly affected by distance (*F*_9,63_ = 6.671; *p* < 0.001; η^2^ = 0.276; Figure 4 Panel C), increasing across the exercise bout, independent of solution. When matched time points were compared, all post-hoc contrasts between solutions were non-significant (all *p* = 1.000). Despite Solution B tending to report higher TS values than menthol-containing solutions, mean differences did not exceed 0.5 arbitrary units, which has previously been considered the smallest worthwhile change [38]. Similarly, TC significantly decreased as distance progressed (*F*_9,63_ = 34.542; *p* < 0.001; η^2^ = 0.682; Figure 4 Panel D), independent of solution. When matched time points were compared, all post-hoc contrasts between solutions were non-significant (all *p* = 1.000).

Peripheral RPE (Legs) was significantly affected by distance completed (*F*_9,63_ = 91.133; *p* < 0.001; η^2^ = 0.832; Figure 4 Panel A). Repeated contrasts showed significant differences between entry and warm up, warm up and 5 km, 30 km and 35 km and 35 km and 40 km (all *p <* 0.05). All repeated contrasts exceeded the threshold of 0.5 arbitrary units. Whilst no significant differences between solutions when matched for distance (all *p* = 1.00) were seen, Solution B was typically ≥0.5 arbitrary units than menthol-containing solutions at matched time points. Central RPE (Lungs) was similarly affected by distance completed (*F*_9,63_ = 66.733; *p* < 0.001; η^2^ = 0.833; Figure 4 Panel B). Repeated contrasts showed significant increases in central RPE between entry and warm up (*p* = 0.047), warm up and 5 km, and 35 km and 40 km (both *p* < 0.001). All repeated contrasts exceeded 0.5 units, except between 5 km and 10 km (range: 0.59 to 2.20); post-hoc testing revealed no consistent trends nor statistically significant differences when matched for distance (all *p* = 1.00).

Thirst showed no significant interaction between distance and solution (*F*_6,42_ = 2.248; *p* = 0.057; η^2^ = 0.093; Figure 4 Panel E); there was a statistically significant within-subjects effect for distance (*F*_3,21_ = 3.662; *p* = 0.029; η^2^ = 0.130), however. Examination of repeated contrasts for distance showed no further significant differences (all *p* > 0.05); however, the menthol solution produced a significantly greater and *moderate* reduction in thirst at 5 km compared to 15 km (0.94; 90% CI 0.38 to 1.50; *p* = 0.007; Figure 4 Panel F), all other comparisons were non-significant.

### 3.4. Location and Intensity of Oral Cooling

There was no significant three-way interaction between distance completed, solution and location upon intensity of oral cooling (*F*_84,588_ = 0.731; *p* = 0.963; η^2^ = 0.095). Further within-subject analyses showed a significant interaction between solution and location upon oral cooling intensity (*F*_28,196_ = 2.577; *p* < 0.001; η^2^ = 0.269), with solution and location exerting significant independent effects (*F*_2,14_ = 6.268; *p* = 0.011; η^2^ = 0.472 and *F*_14,98_ = 12.024; *p* < 0.001; η^2^ = 0.632, respectively). The intensity of oral cooling, for each location, per swill at 5, 15, 25 and 35 km intervals are shown in Figure 5. Broadly, menthol-containing solutions (MEN and BOTH) exerted a greater oral cooling effect compared to carbohydrate swills, across regions associated with the nasal and oral cavities, and these sensations progressed down towards the respiratory tract as exercise duration increased (Figure 5).

## 4. Discussion

This study aimed to investigate the effect of menthol, carbohydrate or combined swilling on 40 km TT performance and resultant physiological and perceptual responses. Contrary to the hypothesis, there was not an additive effect of combining menthol and carbohydrate swills in comparison to employing these strategies in isolation. No significant differences were reported between solutions with respect to 40 km TT performance, with carbohydrate-containing solutions, i.e., CHO and BOTH, considered statistically equivalent to one another. This suggests that during extended self-paced exercise in the heat, fuel availability and associated oral sensing may be a more dominant signal than thermal perception when swills are applied at regular intervals throughout the exercise bout. These findings are contrary to previous research in shorter duration [22,40] and fixed-paced exercise [41,42,43] in the heat, where menthol administration has been shown to be ergogenic, but complement existing carbohydrate swilling literature of similar exercise durations [44,45,46].

As with 40 km TT performance, all trials showed similar mean power outputs (186–188 W) and *large* to *very large* increases in power output in the final 5 km of the trials (Figure 3B). Whilst an end spurt was observed consistently, independent of solution (*p* = 0.903), the lack of statistical equivalence between MEN- and CHO-containing solutions indicates that performance was better maintained throughout the TT by oral sensing of energy availability (CHO and BOTH) than by sensations of oral cooling (MEN). The ergogenic effects of CHO sensing have been shown somewhat consistently in both mechanistic and applied models for over a decade [47,48], especially during glycogen-depleting or fasted exercise [17,49].

Assessment of the intensity and location of cooling is unique to the present study (Figure 2 and Figure 5). We demonstrated that menthol-containing solutions impart a greater oral cooling effect, across regions associated with trigeminal innervation [8,50,51], than CHO swilling, and that this effect is robust across the exercise bout (Figure 5). Some participants also reported a downwards migration of these sensations as exercise progressed; this is not surprising given menthol’s effects are inversely proportional to the thickness of the stratum corneum [51,52,53], and menthol has been shown to increase VE and the drive to breathe [43,54,55], similar to the effects of cold air inhalation [54]. Consequently, migration of sensations of oral cooling is to be expected, but the potential performance-enhancing effect of any accompanying increase in VE from menthol administration may be diminished due to the self-paced nature of the exercise and concomitant lower rate of heat storage [56,57]. Further research into how this sensation may be strategically applied, if at all, is warranted, and preliminary evidence to suggest application at the end of the bout, prior to an end spurt, may be most beneficial [42].

Similar to sensations of oral cooling, menthol elicited significantly *moderate* reductions in thirst early in the exercise bout. This reduction in thirst has been posited by Eccles [54,55] and more recently by Best [58] as a result of menthol’s stimulation of oral cold receptors and refreshing sensation. The *small* differences in fluid intake between menthol-containing solutions and CHO indicate that this hypothesis is supported by our data. Similar effects are noted as a result of the ‘bite’ individuals experience when drinking carbonated water, which is considered more thirst satiating than still water because of its CO_2_ content [59,60]. This reduction in thirst, however, did not appear to alter performance or mean power output, suggesting that as athletes progress through exercise in the heat hyperthermia and associated perceptual responses likely outweigh modifications in thirst brought about by menthol mouth swilling, as per Figure 4F.

In contrast to previous research [9], we demonstrated no meaningful alterations in thermal sensation as a result of administering menthol. There are several potential reasons for this. Firstly, we included a source of convective cooling that is often lacking or minimal in closed environmental testing. Researchers may often minimise sources of convective cooling, to maximise the heat storage response experienced by athletes and thus increase the likelihood of a perceptual cooling effect, at the potential expense of ecological validity [61,62]. When this convective cooling is combined with a liquid swill, irrespective of composition, the stimulation of oral hygro-receptors is sufficient to mitigate increases in thermal sensation, as per Eccles’ model of oral cooling [55]. Potential decreases in participants’ thermal sensation as a result of (menthol) swilling may have been countered by the radiant heat load. This was intentionally positioned across participants’ backs to stimulate a large, heat-sensitive area [63] and further replicate a real-world competitive environment. Finally, participants may consider thermal perceptual signals as gestalt or full-body interpretations of the factors listed above, further paired with metabolic heat production from exercise. Given the ability of menthol to exert significant and localised oral cooling effects, we would encourage researchers and practitioners to consider moving towards a differential approach based upon sensory inputs of interest when assessing thermal comfort and/or sensation. Similar approaches have been used with ratings of perceived exertion [64] and permit the examination of multiple (perceptual) inputs under differing contexts, e.g., playing position or environmental conditions [65,66]. Adopting a similar approach may be useful for environmental physiologists, practitioners and athletes wishing to employ perceptual and physiological cooling strategies either concurrently or intermittently throughout an exercise bout, training cycle or competition.

The above also suggests a need to repeat this investigation or similar in heat-acclimated individuals. The benefits are a lower coefficient of variation in performance and physiological responses, as well as a stabilisation of perceptual measures [67,68] between trials, potentially increasing reliability of participant responses and allowing examination of smaller meaningful effect sizes, with regard to alterations in perceptual and physiological measures.

The absence of a water-only swill is a mechanistic weakness of the present study, as it does not allow the methodological isolation of the effect of (non-nutritive) swilling alone compared to swills that may impart nutritive and or perceptual effects. This is partially mitigated by using flavourless carbohydrate in the CHO and BOTH solutions. There is also the question of whether cyclists do or would simply drink water alone during competitive efforts, especially in the heat. CHO provision is strongly encouraged by nutrition guidelines [69,70], and questionnaire data from cycling and ironman events demonstrate CHO consumption of between 30 and 90 g.h^−1^ [69,71,72] during competition. Given the seemingly widespread use of CHO solutions compared to water, the mechanistic sacrifice appears to be outweighed by furthering the ecological validity of the research question.

Translating the current findings into practice, the requirement to move out of the aerodynamic position and potentially modify breathing patterns to undertake the swill may impact negatively on performance and negate any added benefits gained by the swill [73,74]. Gam et al. noted that the act of non-nutritive rinsing (water), compared to no rinsing, incurred a performance cost that could be mitigated by CHO swilling [74]; hence, the ergogenic effect of the swill must outweigh the detrimental effect of swilling. Thus, in exercise settings (duration and intensity dependent) where performance may not be limited by glycogen depletion, hyperthermia or thermal or thirst sensations, we would argue that the maintenance of an aerodynamic position likely confers a greater ergogenic effect than that provided by a swill. Swilling could, however, be utilised in longer or indoor heat acclimation training sessions [75] or during longer ultra-events in the heat (e.g., world ironman championships [76]), where there is a large and repetitive fuelling burden, and the accompanying perceptual cooling may also serve to disrupt flavour fatigue and possibly minimise upper gastrointestinal symptoms by providing a contrasting taste and targeting different oral receptors.

Familiarisation is a final and important consideration when interpreting the results from this paper against the body of literature to date and when conducting future work. A similarly designed, 40 km TT study in the heat [20] that compared placebo to CHO mouth swill found no significant difference in all recorded measures, echoing our findings. There was, however, no familiarisation prior to experimental sessions. Whilst not entirely uniform (i.e., not statistically significant), *moderate* differences between familiarisation and experimental sessions were observed in the present study, indicating that the extra trial is of value to participants, and familiarisation data should be reported by researchers. We demonstrated a similar effect in menthol mouth rinsing during strength- and power-based activity under normothermic conditions [26]. It is important to note that participants were encouraged to commit fully to the familiarisation trial and adopt a pacing strategy that allowed them to work as hard as possible, for as long as possible, with minimal external feedback.

## 5. Conclusions

In conclusion, when menthol and carbohydrate mouth rinses were combined, no additive effect was observed. Menthol imparted greater sensations of oral cooling and attenuated thirst to a greater extent, especially earlier in the exercise bout. Carbohydrate-containing swills (CHO and BOTH) were statistically equivalent to each other but were not equivalent to menthol swilling (i.e., menthol was slower). This suggests that during glycogen-depleting exercise in the heat, oral sensing of carbohydrate, and thus energy availability and subsequent signalling, likely confers a greater advantage than a localised perceptual cooling response. Practically, it is inferred that swilling may incur an aerodynamic cost during competition but may be a useful nutritional strategy during prolonged or hot training sessions.

## Figures and Tables

**Figure 1 nutrients-13-04309-f001:**
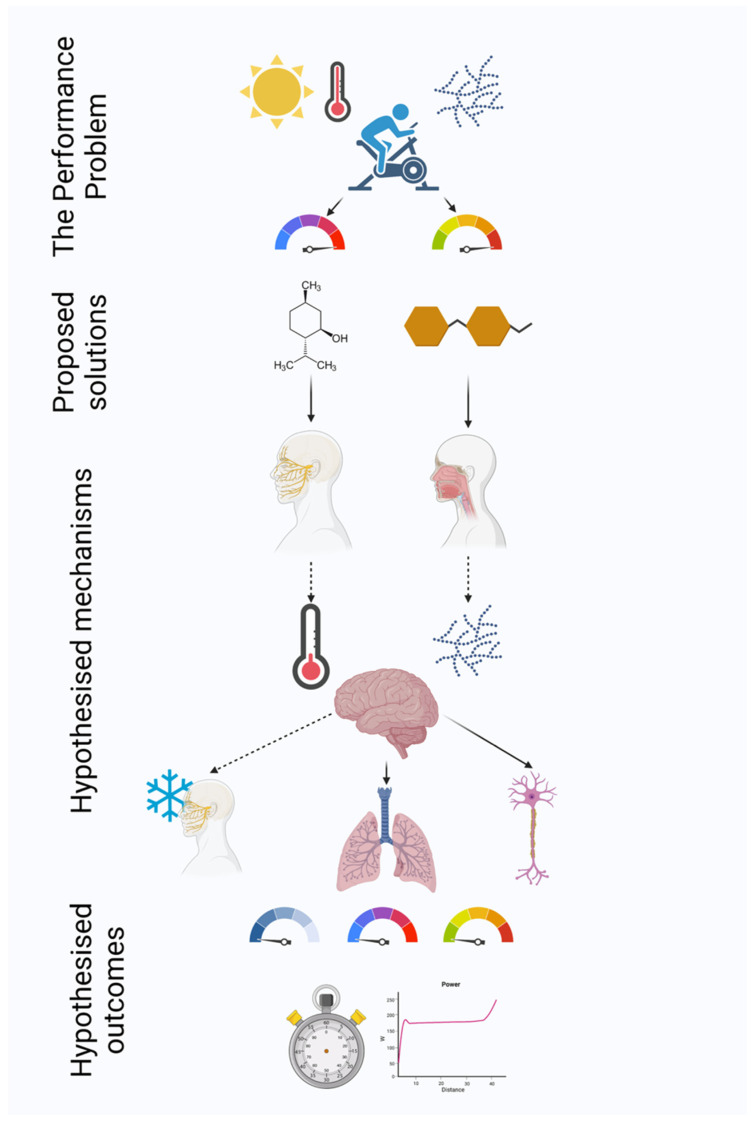
The performance of endurance exercise in the heat with solar radiation imposes considerable glycogen depletion, with the environmental conditions serving to increase thermal sensation(s) and ratings of perceived exertion (the performance problem). Proposed solutions to this are menthol and carbohydrate administration or a combination of both solutions. When swilled, these solutions stimulate oral cold receptors and the trigeminal nerve and oral carbohydrate receptors, respectively. In turn, this may lower perceived temperature (menthol) and increase perceived fuel availability (carbohydrate). The brain may then interpret these signals, imparting a localised cooling effect, altering ventilatory mechanics or increasing central drive. These mechanisms may then improve perceptions of local cooling, full-body thermal sensation or RPE (left to right dials), translating to improvements in time trial performance via maintenance of a greater power output during exercise (hypothesised outcomes). Dashed arrows indicate perceptual drivers, whereas solid arrows indicate objectively measurable drivers. Created with BioRender.com (accessed on 30 September 2021).

**Figure 2 nutrients-13-04309-f002:**
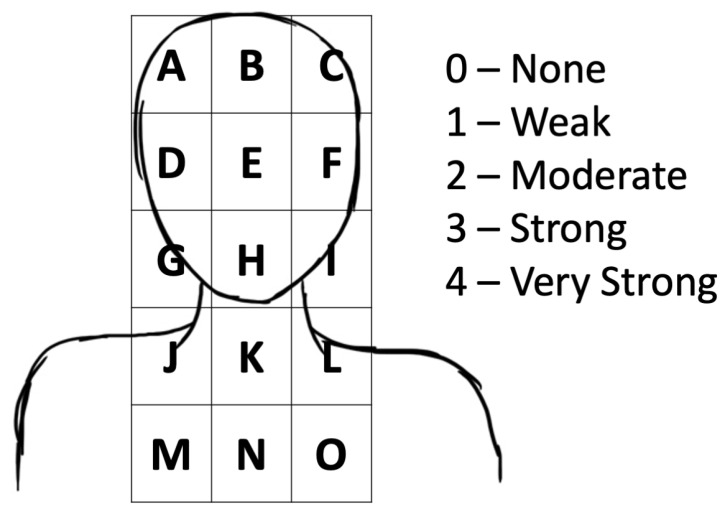
Novel oral cooling scale. Used to rate the location and intensity of (oral) cooling across regions that are likely to be menthol sensitive.

**Figure 3 nutrients-13-04309-f003:**
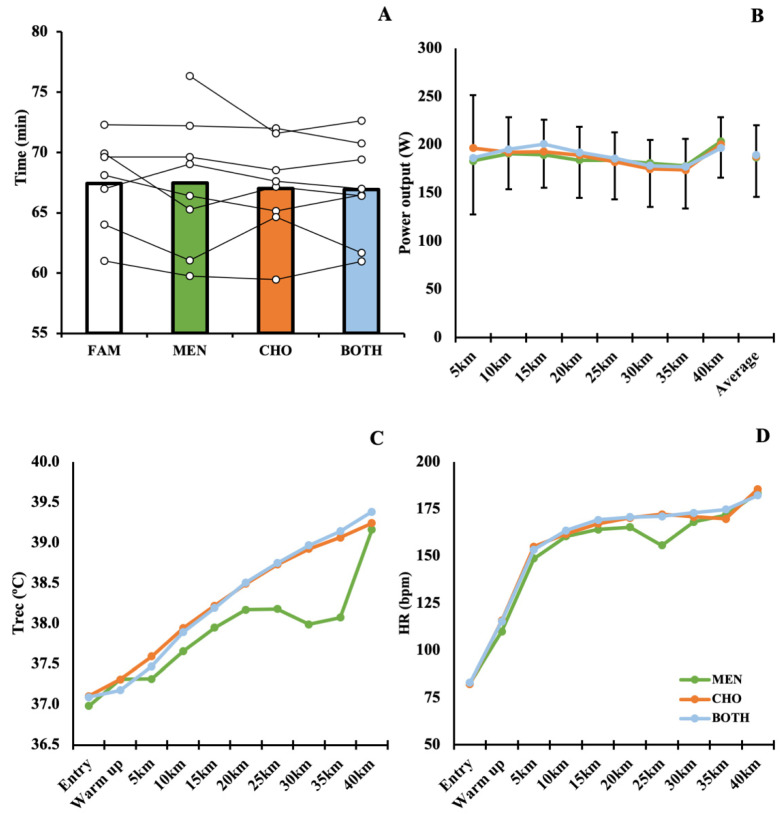
Mean and individual time trial performances (**A**) and mean power output (**B**) per condition. Familiarisation (**FAM**) is shown in panel (**A**) to aid in interpretation of results and support subsequent discussion. T_rec_ (**C**) and HR (**D**) responses are presented as mean values, for each time point. Error bars are not shown in panels (**C**,**D**), to improve visibility of the data.

**Figure 4 nutrients-13-04309-f004:**
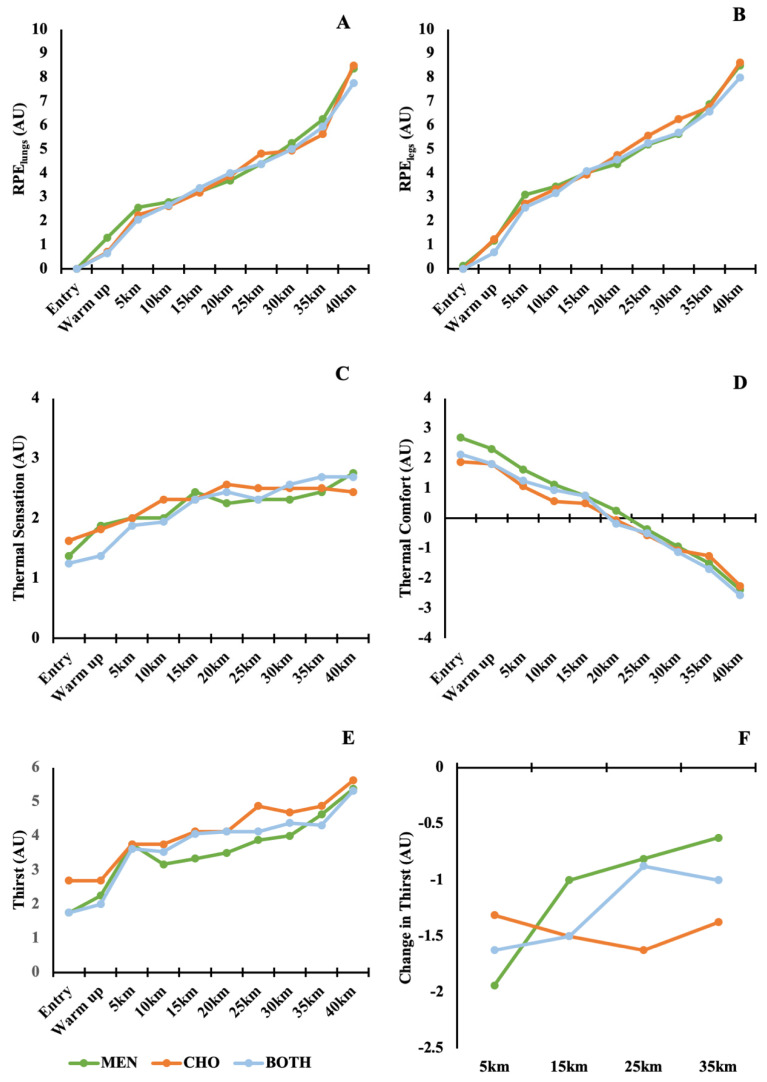
Perceptual responses to each swilling intervention. Panels show central (**A**) and peripheral (**B**) RPE, TS (**C**) and TC (**D**), thirst (**E**) and change in thirst (**F**). Data shown are mean responses; error bars are not presented to improve visibility of the data.

**Figure 5 nutrients-13-04309-f005:**
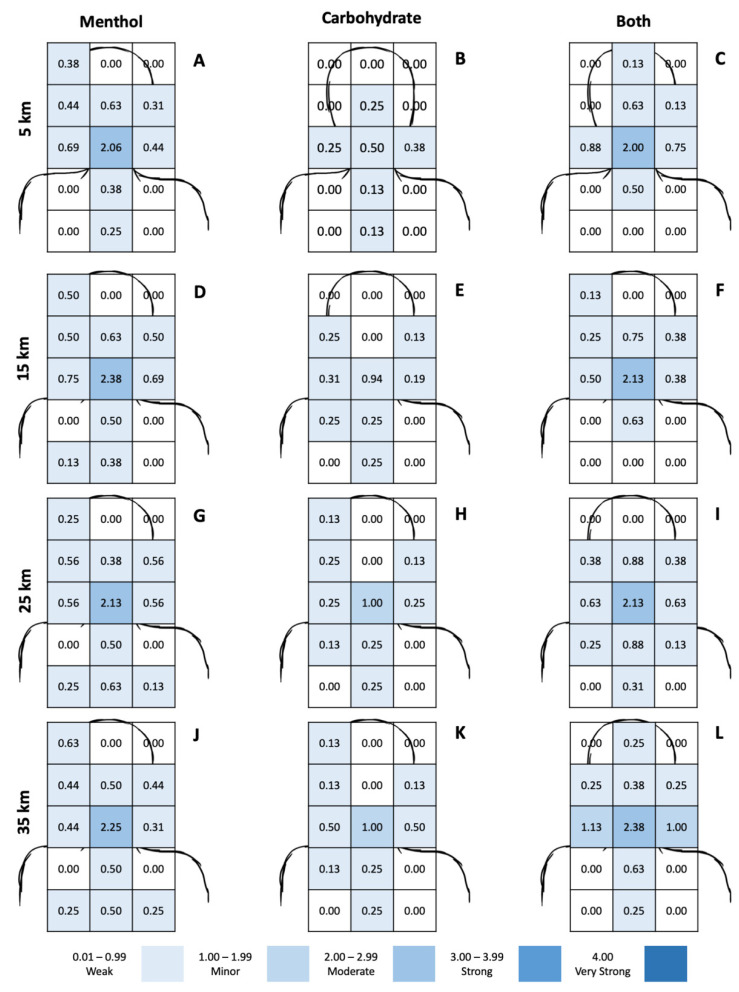
Location and intensity of oral cooling, by differing solutions. Columns are assigned to differing solutions: Panels (**A**,**D**,**G**,**J**) are for MEN; panels (**B**,**E**,**H**,**K**) are for CHO; panels (**C**,**F**,**I**,**L**) are for BOTH. Rows represent time points, 5 km (**A**–**C**), 15 km (**D**–**F**), 25 km (**G**–**I**) and 35 km (**J**–**L**) from top to bottom, respectively.

## Data Availability

The data presented in this study are available on request from the corresponding author. The data are not publicly available due to institutional data policies, however an anonymised, secure copy will be stored on Researchgate.

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
