# Peer review of "The Effect of Isolated and Combined Application of Menthol and Carbohydrate Mouth Rinses on 40 km Time Trial Performance, Physiological and Perceptual Measures in the Heat"

_nutrients, 2021, doi:10.3390/nu13124309_

Round 1

Reviewer 1 Report

Thank you for allowing me to read this interesting and well-written manuscript. I do feel that the authors should consider using mouth ‘rinse / rinsing’ rather than ‘swills’ in order to be consistent with the bulk of the previously published studies. Otherwise, I have just some minor points to consider.

Introduction:

Lines 67-69: It would be worth highlighting that CHO mouth rinse improved performance to a greater extent in a fasted compared with a fed state; however, optimal performance was achieved in a fed state with the addition of a CHO mouth rinse.

Lines 79-80: A reference to support this would be helpful.

A really helpful summary is presented in figure 1.

Methods:

Report age as an integer, and in the abstract.

Statistical analysis:  A justification of the sample size would be beneficial.

Results:

Helpful inclusion of individual responses. Was there any order effect?

Discussion:

A nice discussion combining the results and the potential underlying mechanisms. However, I do feel that it is important to highlight the potential limitation of no control condition. Gam et al. (2013) reported the act of rinsing the mouth during a cycle time trial had a detrimental effect on performance, although the addition of carbohydrate to the rinse solution reduced the decrease in performance associated with repeated mouth rinsing. Therefore, it is possible that the act of rinsing the mouth during the time trials caused a loss of attention and focus on the task resulting in these transient declines in performance.

I hope that the authors find the above comments helpful and in the constructive manner they are intended.

Author Response

Thank you for allowing me to read this interesting and well-written manuscript. I do feel that the authors should consider using mouth ‘rinse / rinsing’ rather than ‘swills’ in order to be consistent with the bulk of the previously published studies. Otherwise, I have just some minor points to consider.

Thank you for the compliment regarding the manuscript. It is a really refreshing opener to a review and the team really appreciates it. We have amended to rinsing for consistency, as suggested, and have tended to the other minor considerations raised below.

Introduction:

Lines 67-69: It would be worth highlighting that CHO mouth rinse improved performance to a greater extent in a fasted compared with a fed state; however, optimal performance was achieved in a fed state with the addition of a CHO mouth rinse.

Thank you for this suggestion, we have added some further comment to clarify this. This section now includes 'Performance is typically improved to a greater extent when conducted in a fasted state, but would still be considered nutritionally optimized in the fed-state.' 

Lines 79-80: A reference to support this would be helpful.

Thank you for this suggestion, we have added a reference to support this.

A really helpful summary presented in figure 1.

Methods:

Report age as an integer, and in the abstract.

Amended as requested.

Statistical analysis:  A justification of the sample size would be beneficial.

Thank you for this suggestion. We have referenced an article by Daniel Lakens to this effect, who notes that there are valid reasons beyond statistical power as to why a sample size is chosen. We were constrained by resources, athlete availability and pandemic protocols. This is stated and the article referenced on lines 282-284. We appreciate this is not ideal, but prefer to be transparent.

Results:

Helpful inclusion of individual responses. Was there any order effect?

Thank you for this comment, there was no order effect apparent in the data for any of the main parameters of interest. 

Discussion:

A nice discussion combining the results and the potential underlying mechanisms. However, I do feel that it is important to highlight the potential limitation of no control condition. Gam et al. (2013) reported the act of rinsing the mouth during a cycle time trial had a detrimental effect on performance, although the addition of carbohydrate to the rinse solution reduced the decrease in performance associated with repeated mouth rinsing. Therefore, it is possible that the act of rinsing the mouth during the time trials caused a loss of attention and focus on the task resulting in these transient declines in performance.

Thank you for this suggestion, we have included the reference from Gam et al., and this section now reads as follows:

'Translating the current findings into practice, the requirement to move out of the aerodynamic position and potentially modify breathing patterns to undertake the swill may impact negatively on performance and negate any added benefits gained by the swill [73,74]. Gam et al., noted that the act of non-nutritive rinsing (water), compared to no rinsing, incurred a performance cost that could be mitigated by CHO swilling [74]; hence, the ergogenic effect of the swill, must outweigh the detrimental effect of swilling. Thus, in exercise settings (duration and intensity dependent) where performance may not be limited by glycogen depletion, hyperthermia or thermal or thirst sensations, we would argue that the maintenance of an aerodynamic position likely confers a greater ergogenic effect than that provided by a swill. '

I hope that the authors find the above comments helpful and in the constructive manner they are intended.

Thank you, we really appreciated the comments and the constructive nature of the feedback provided. We hope that the attended changes are satisfactory, and again would like to thank the reviewer for comments that have improved the manuscript.

Reviewer 2 Report

Although many studies have been conducted on the effects of mouse rinses on athletic performance, there is insufficient knowledge on the effects of mouse rinses in hot environments. The authors conducted a human study to investigate the effects of carbohydrate, menthol, and mixed solution ingestion on 40 km time trial performance in a hot environment.

  1. As a whole, half-width characters are used between numbers and units. Some are left blank and some are not, so they should be standardized to be left blank throughout the paper. However, % and ˚C are exceptions.
  2. Use “mL” instead of “ml”, please.
  3. Line 14: Delete “(8)”, please.
  4. Line 16: “BOTH” is already defined in line 11, so “a combitation (BOTH)” should be “BOTH”.
  5. Line 74: comma after "heat” can be removed.
  6. Method: The gender of the subject should be described.
  7. Line 108: What are (-0.33) and (-0.71)? Please describe.
  8. Line 117: Please cite references for this measurement.
  9. Line 125: The underline of “and” is unnecessary.
  10. Line 144: Please describe the washout period between experimental trials.
  11. Line 160: State the manufacturer and product name of maltodextrin.
  12. Line 179 and 183: Remove the underline under “45˚”.

Line 197: Please describe the references for this method. It is unclear from the chart which part of the mouth the recording form corresponds to.

  1. Line 202: Please list the total fluid intake in the results section.
  2. Line 258: Please list the sweat rate in the results section.
  3. Figure 2 and 3: Tick is missing from the X-axis.
  4. Figure 2C: Why did the Trec decrease between 20-35 km and return at 40 km?
  5. Line 289-307: Are you asking about RPE by site? Please describe in the method.
  6. Figure 3: The unit of Y-axis is arbitrary unit. but you did not experiment with VAS scaled?
  7. Figure 3F: Unfortunately I don't understand the necessity of this graph, could you please delete it?
  8. Line 324: There is a lack of explanation in the Figure 4. For example, does A, D, G, and J correspond to 5, 15, 25, and 35 km, respectively?

Author Response

Although many studies have been conducted on the effects of mouse rinses on athletic performance, there is insufficient knowledge on the effects of mouse rinses in hot environments. The authors conducted a human study to investigate the effects of carbohydrate, menthol, and mixed solution ingestion on 40 km time trial performance in a hot environment.

Thank you for the comments provided and recommended revisions, we have considered and tended to them all, and have added further explanation as appropriate.

  1. As a whole, half-width characters are used between numbers and units. Some are left blank and some are not, so they should be standardized to be left blank throughout the paper. However, % and ˚C are exceptions.

Amended as requested

  1. Use “mL” instead of “ml”, please.

Amended as requested

  1. Line 14: Delete “(8)”, please.

Amended as requested

  1. Line 16: “BOTH” is already defined in line 11, so “a combitation (BOTH)” should be “BOTH”.

Amended as requested

  1. Line 74: comma after "heat” can be removed.

Amended as requested

  1. Method: The gender of the subject should be described.

Amended as requested

  1. Line 108: What are (-0.33) and (-0.71)? Please describe.

Thank you for this suggestion, we have included the unit d to confirm that this is Cohen's d, as a measure of effect size.

  1. Line 117: Please cite references for this measurement.

Thank you for this suggestion, this is a novel scale, as detailed in the methods later, so cannot be referenced at this time.

  1. Line 125: The underline of “and” is unnecessary.

Amended as requested

  1. Line 144: Please describe the washout period between experimental trials.

Amended as requested

  1. Line 160: State the manufacturer and product name of maltodextrin.

Amended as requested

  1. Line 179 and 183: Remove the underline under “45˚”.

Amended as requested

Line 197: Please describe the references for this method. It is unclear from the chart which part of the mouth the recording form corresponds to.

A blank version of the chart has been added as a figure below this section of the methods (Figure 2)

  1. Line 202: Please list the total fluid intake in the results section.

Amended as requested

  1. Line 258: Please list the sweat rate in the results section.

Amended as requested

  1. Figure 2 and 3: Tick is missing from the X-axis.

Amended as requested

  1. Figure 2C: Why did the Trec decrease between 20-35 km and return at 40 km?

Thank you for this comment. This may be due to a pacing effect plus an end spurt, and or an artefact within the data. There is no physiological explanation, so we have elected not to comment upon these data, as we do not think they would be replicated if someone else were to undertake the same investigation again.

  1. Line 289-307: Are you asking about RPE by site? Please describe in the method.

Thank you for this comment, yes we did document RPE by site (peripheral and central/ legs and lungs). This is reported in the methods at lines 260-262.

  1. Figure 3: The unit of Y-axis is arbitrary unit. but you did not experiment with VAS scaled?

Our understanding of an arbitrary unit is a unit that scales to a particular sensation or event within an individual. Whilst this trend may be experienced similarly between individuals, e.g. an increase in thermal sensation with increasing environmental temperature, each individual has a different experience so the unit of measurement is considered arbitrary. Similar examples may be pain or anxiety which are rated on a VAS or Likert scale, but are unitless/dimensionless.

  1. Figure 3F: Unfortunately I don't understand the necessity of this graph, could you please delete it?

Thank you for this suggestion. We wish to include this graph because it details the change in thirst as a result of swilling at each time point. These data are important when considered in conjunction with the fluid intake reported per condition, as they demonstrate menthol's ability to reduce thirst, independent of an intake of fluid.

  1. Line 324: There is a lack of explanation in the Figure 4. For example, does A, D, G, and J correspond to 5, 15, 25, and 35 km, respectively?

We have amended the caption to this effect, and included solution and distance headers for each column and row, respectively. We hope this is clear now.